# SQuant: On-the-Fly Data-Free Quantization via Diagonal Hessian Approximation

**Cong Guo**[1,2], **Yuxian Qiu**[1,2], **Jingwen Leng**[1,2, *], **Xiaotian Gao**[3], **Chen Zhang**[4], **Yunxin Liu**[5], **Fan Yang**[3], **Yuhao Zhu**[6] **& Minyi Guo**[1,2, *]

[1] Shanghai Jiao Tong University, [2] Shanghai Qi Zhi Institute
[3] Microsoft Research, [4] DAMO Academy, Alibaba Group
[5] Institute for AI Industry Research (AIR), Tsinghua University, [6] University of Rochester
`{guocong, qiuyuxian, leng-jw}@sjtu.edu.cn`
`{xiaotian.gao, fanyang}@microsoft.com`
`mingchong.zc@alibaba-inc.com, liuyunxin@air.tsinghua.edu.cn`
`yzhu@rochester.edu, guo-my@cs.sjtu.edu.cn`

## Abstract

Quantization of deep neural networks (DNN) has been proven effective for compressing and accelerating DNN models. Data-free quantization (DFQ) is a promising approach without the original datasets under privacy-sensitive and confidential scenarios. However, current DFQ solutions degrade accuracy, need synthetic data to calibrate networks, and are time-consuming and costly. This paper proposes an on-the-fly DFQ framework with sub-second quantization time, called SQuant, which can quantize networks on inference-only devices with low computation and memory requirements. With the theoretical analysis of the second-order information of DNN task loss, we decompose and approximate the Hessian-based optimization objective into three diagonal sub-items, which have different areas corresponding to three dimensions of weight tensor: element-wise, kernel-wise, and output channel-wise. Then, we progressively compose sub-items and propose a novel data-free optimization objective in the discrete domain, minimizing Constrained Absolute Sum of Error (or CASE in short), which surprisingly does not need any dataset and is even not aware of network architecture. We also design an efficient algorithm without back-propagation to further reduce the computation complexity of the objective solver. Finally, without fine-tuning and synthetic datasets, SQuant accelerates the data-free quantization process to a sub-second level with $> 30\%$ accuracy improvement over the existing data-free post-training quantization works, with the evaluated models under 4-bit quantization. We have open-sourced the SQuant framework[1].

## 1 Introduction

With the widespread application of DNN, more and more DNN models are deployed on both computation-constrained and memory-constrained environments, e.g., smartphones, IoT devices, and self-driving cars. The desire for lightweight and energy-efficient DNN deployment solutions is increasing. Quantization is one of the most promising techniques to convert weights and activations to lower bit formats and simultaneously reduce computational time and memory consumption. There are two kinds of quantization: Post-training quantization (PTQ) (Banner et al., 2018; Choukroun et al., 2019; Zhao et al., 2019; Nagel et al., 2020) and Quantization-aware training (QAT) (Gupta et al., 2015; Jacob et al., 2018; Wang et al., 2019; Zhuang et al., 2021). QAT requires to simulate quantization in the training process, which invokes time-consuming retraining and hyper-parameter tuning. In contrast, PTQ directly quantizes well-trained models without retraining. However, they still need training datasets to calibrate (Nagel et al., 2020) quantized models but are often unavailable due to privacy and security issues, such as medical and confidential scenarios.

---

*Jingwen Leng and Minyi Guo are corresponding authors of this paper.
[1] https://github.com/clevercool/SQuant

In contrast, data-free quantization (DFQ) has recently been presented as a promising way to quantize models without original datasets (Nagel et al., 2019; Cai et al., 2020; Zhang et al., 2021; Xu et al., 2020; Liu et al., 2021; Qin et al., 2021; Choi et al., 2020). From a deployment perspective, DFQ is the most attractive quantization method since we can apply it to any trained models as a black box post-processing step. However, current DFQ methods cannot achieve high accuracy and fast processing time simultaneously. Traditionally, DFQ (Nagel et al., 2019) uses rounding quantization, leading to the rounding-to-nearest strategy. Such a strategy causes significant accuracy loss, especially in low-bit settings. To bridge the accuracy gap between data-free and data-driven quantization, researchers propose a series of data-generative DFQ methods. They use gradient-based methods to generate fake datasets for trained models. With the synthetic data, they can employ a data-driven calibration and fine-tuning strategy to improve accuracy. However, data generation typically adopts the time-consuming gradient-based methods, which require multiple iterations to generate each input. For example, prior works often spend hours generating a calibration dataset and fine-tuning the network (Xu et al., 2020; Liu et al., 2021; Zhang et al., 2021).

To solve this dilemma, we propose SQuant, a fast and accurate data-free quantization framework for convolutional neural networks, employing the constrained absolute sum of error (CASE) of weights as the rounding metric. By leveraging Hessian information of network loss due to quantization, we propose a novel diagonal Hessian approximation, which decomposes the optimization objective into three data-free sub-items: element-wise, kernel-wise, and output channel-wise, each of which corresponds to a single or a set of dimensions of the weight tensor. We progressively compose and optimize these three sub-items in the discrete space. The final approximate objective eliminates the requirement of data generation. We propose a progressive algorithm with linear complexity to solve the optimization objective, further accelerating DFQ time to a sub-second level. For example, SQuant only needs an average of 4 ms and 84 ms for quantizing a layer and the overall network of ResNet18, respectively. As it does not require back-propagation nor fine-tuning, SQuant can run on inference-only devices with limited computation and memory resources on the fly. That opens up new opportunities and scenarios for adopting quantization.

Compared with state-of-the-art DFQ methods, SQuant achieves higher accuracy on all evaluated models under the 4/6/8-bit settings. SQuant only introduces 0.1% accuracy loss on average under the 8-bit setting. Under fewer bit precisions, the advantage of SQuant further expands. SQuant only introduces 1.8% accuracy loss on average under the 6-bit setting. Under the 4-bit setting, SQuant can achieve more than 30% accuracy improvement compared with data-free PTQ methods. In a word, SQuant pushes the accuracy and processing time of DFQ to a new frontier.

## 2 PRELIMINARIES

### 2.1 NOTATIONS

We specifically use $x$, $y$ and $w$ to denote the input, output, and weight variables, respectively. Constant and scalar are denoted by italic letters, e.g., $c, M$. Column vector and flattened matrix are denoted by bold lowercase letters, e.g., $\mathbf{w}$, and matrices (or tensors) are represented by uppercase letters, e.g., $\mathbf{W}$. The subscript and superscript can further represent the element indices and the layer of a network, respectively, e.g., $\mathbf{W}_{i,j}^{\ell}$. $\mathbb{E}[\cdot]$ denotes the expectation operator, and the network loss function is represented by $\mathscr{L}(\cdot)$. For convenience in this paper, we call the row of FC (fully connected layer) weight as the output channel and the column of FC weight as the input channel, which are the counterparts to Conv (convolution layer) weight. We use $M$, $N$, and $K$ to denote output channel size, input channel size, and kernel height $\times$ kernel width, respectively. Specifically, FC has the shape of $(M, N, 1)$.

### 2.2 QUANTIZATION

Most previous works adopt the rounding-to-nearest approach for quantizing deep neural networks by rounding elements $w$ to the nearest quantization grid values with a fixed-point data type. The quantization and dequantization for a quantized element $\widehat{w}$ can be described as $\widehat{w} = s \cdot \mathrm{clip}(\lfloor \frac{w}{s} \rceil, min, max)$, where $s$ denotes the quantization scale parameter and, $min$ and $max$ are the lower and upper thresholds for the clipping function $\mathrm{clip}(\cdot)$. The operator $\lfloor \cdot \rceil$ represents the rounding-to-nearest, i.e., minimizing the mean squared error (MSE) between the quantized and the original value.

## 2.3 Hessian-Based Optimization for Neural Networks

The Hessian-based approach is one of the most promising optimizations to further improve the quantization (Dong et al., 2019b;a; Nagel et al., 2020; Shen et al., 2020; Qian et al., 2020; Wu et al., 2020; Hubara et al., 2020; Li et al., 2021; Yao et al., 2021) and pruning (Yu et al., 2021) performance for DNN models. Some of those works exploit the Hessian matrix to approximate loss degradation due to the quantization perturbation of weight, $\Delta \mathbf{W}$, by

$$\mathbb{E}[\mathscr{L}(\mathbf{X}, \mathbf{Y}, \mathbf{W} + \Delta \mathbf{W}) - \mathscr{L}(\mathbf{X}, \mathbf{Y}, \mathbf{W})] \approx \mathbb{E}[\Delta \mathbf{W} \cdot \mathbf{g}^{\mathbf{W}} + \frac{1}{2}\Delta \mathbf{W} \cdot \mathbf{H}^{\mathbf{W}} \cdot \Delta \mathbf{W}^T], \quad (1)$$

where the equation comes from second-order Taylor series expansion, $\mathbf{g}^{\mathbf{W}}$ is the gradient and $\mathbf{H}^{\mathbf{W}}$ is the full network Hessian matrix w.r.t. original weight, $\mathbf{W}$. Since a well-trained model has already converged, the gradient term will be close to 0 and thus can be safely ignored. However, computing $\mathbf{H}^{\mathbf{W}}$ is infeasible because of the large memory overhead and computation complexity. To tackle this problem, we approximate $\mathbf{H}^{\mathbf{W}}$ as a layer-wise Hessian matrix $\mathbf{H}^{\mathbf{W}^\ell}$ under the assumption of cross-layer independence (Dong et al., 2017; Nagel et al., 2020), i.e., $\mathbf{H}^{\mathbf{W}^\ell} = \mathbf{x}^\ell \mathbf{x}^{\ell T} \otimes \nabla^2_{\mathbf{y}^\ell}\mathscr{L}$, where $\otimes$ denotes Kronecker product of two matrices, $\nabla^2_{\mathbf{y}^\ell}\mathscr{L}$ is the Hessian of the task loss w.r.t. $\mathbf{y}^\ell$.

For the $m$-th output channel of Conv or FC, $\mathbf{H}^{\mathbf{W}^\ell}$ can be approximatively simplified into output channel-wise (Nagel et al., 2020; Yu et al., 2021; Wu et al., 2020; Qian et al., 2020),

$$\mathbf{H}^{\mathbf{W}^\ell_m} \approx \nabla^2_{\mathbf{y}^\ell}\mathscr{L}_{m,m} \cdot \mathbf{x}^\ell \mathbf{x}^{\ell T} = l_m \cdot \mathbf{x}^\ell \mathbf{x}^{\ell T}, \quad (2)$$

where $\nabla^2_{\mathbf{y}^\ell}\mathscr{L}$ is approximately a diagonal matrix. Then the final optimization objective is

$$\Delta \widehat{\mathbf{W}}^\ell_{m,:} = \underset{\Delta \mathbf{W}^\ell_{m,:}}{\arg\min} \quad \Delta \mathbf{W}^\ell_{m,:} \mathbb{E}[\mathbf{H}^{\mathbf{W}^\ell_m}] \Delta \mathbf{W}^\ell_{m,:}{}^T \quad (3)$$

$$\approx \underset{\Delta \mathbf{W}^\ell_{m,:}}{\arg\min} \quad \Delta \mathbf{W}^\ell_{m,:} \mathbb{E}[\mathbf{x}^\ell \mathbf{x}^{\ell T}] \Delta \mathbf{W}^\ell_{m,:}{}^T = \underset{\Delta \mathbf{W}^\ell_{m,:}}{\arg\min} \mathbb{E}[(\Delta \mathbf{W}^\ell_{m,:} \mathbf{x}^\ell)^2], \quad (4)$$

which is the MSE between the output activation produced from original and quantized weights. Each sub-problem deals with a single output channel $\Delta \mathbf{W}^\ell_{m,:}$. We will further approximate Eq. (4) to remove any input data dependency from the optimization objective in Sec. 3.2.

# 3 Methodology

## 3.1 Overview

Although we can obtain a good quantization strategy by minimizing MSE for each output channel, it is an NP-hard combinatorial optimization problem. Even approaching an acceptable local minimum requires significant effort and involves input activations without the data-free promise.

To avoid the combinatorial optimization problem and eliminate the requirement of data, we propose the **SQuant** framework. First, SQuant approximates Eq. (4) with three diagonal Hessian matrices corresponding to the dimensions of weight, in Sec. 3.2. Due to the quantization with a fixed-point data type, SQuant transforms the problem into a data-free optimization problem in the discrete domain. SQuant dedicates to optimizing each layer's weight employing a flipping approach (Nagel et al., 2020) without any input activation. To achieve our proposed optimization objective, minimizing CASE (Constrained Absolute Sum of Error), SQuant progressively composes three approximate sub-items under constraint relaxation, introduced in Sec. 3.3. Finally, SQuant needs to work out a flipping set $\mathbf{f}$ to minimize the CASE of each kernel and output channel. We design an efficient algorithm with a linear computation complexity to find a proper $\mathbf{f}$ based on Eq. (8), in Sec. 3.4.

## 3.2 Diagonal Hessian Approximation

In this work, we propose a new approximation of the Hessian matrix to cover non-diagonal elements and decompose Eq. (4) into three sub-items that correspond to the three dimensions of the weight

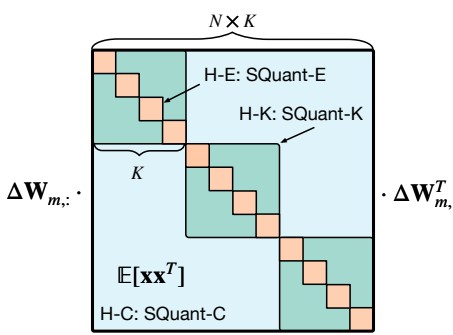

Figure 1: $\Delta \mathbf{W}_{m,:} \mathbb{E}[\mathbf{x} \mathbf{x}^T] \Delta \mathbf{W}_{m,:}^T$. SQaunt-E, SQaunt-K, and SQuant-C are three approximate sub-items, which cover `H-E`, `H-K` and `H-C`, respectively.

tensor as illustrated in Fig. 1: **SQuant-E** for element-wise optimization covers the diagonal elements of $\mathbf{H}^{\mathbf{w}_m^\ell}$ (`H-E`); **SQuant-K** for kernel-wise optimization covers the diagonal blocks of $\mathbf{H}^{\mathbf{w}_m^\ell}$ (`H-K`); **SQuant-C** for output channel-wise optimization covers the whole $\mathbf{H}^{\mathbf{w}_m^\ell}$ (`H-C`).

The $\mathbb{E}[\mathbf{x}^\ell \mathbf{x}^{\ell T}]$ can be approximated by the following equation:

$$\mathbb{E}[\mathbf{x}^\ell \mathbf{x}^{\ell T}] \approx \mathbf{E} + \mathbf{K} + \mathbf{C}, \tag{5}$$

where $\mathbf{C} = c_m \mathbf{J}_{NK}$,

$$\mathbf{K} = \begin{bmatrix} k_1 \mathbf{J}_K & & \\ & \ddots & \\ & & k_N \mathbf{J}_K \end{bmatrix}, \text{ and } \mathbf{E} = \begin{bmatrix} e_{1,1} & & \\ & \ddots & \\ & & e_{N,K} \end{bmatrix}.$$

In the above equations, $\mathbf{J}_{NK}$ is an all-one matrix with the dimension of $N \times K$ (denoted as $NK$), and $c_m$ is a constant value for $m$-th output channel. $\mathbf{K}$ is a diagonal block matrix, where $\mathbf{J}_K$ represents an all-one matrix with the dimension of $K \times K$. The $n$-th diagonal block corresponds to $n$-th kernel in convolution and has its own constant value $k_n$. $\mathbf{E}$ is a diagonal matrix with the diagonal elements of $e_{n,i}$, each of which is a constant value corresponding to $i$-th element of $n$-th kernel.

Eq. (5) provides an approximation that preserves as much information from three different levels of $\mathbb{E}[\mathbf{x}^\ell \mathbf{x}^{\ell T}]$ as possible, which we explain in Appendix A.1. The matrix $\mathbf{C}$ catches the common component of the Hessian matrix, while the matrix $\mathbf{E}$ reserves the individual components in the diagonal line of the Hessian matrix. In addition, we consider kernel-wise approximation for convolution layers by using matrix $\mathbf{K}$. For each inference, the weights of a kernel, $\mathbf{W}_{m,n,:}^\ell$, scan the same feature map. As a result, the corresponding $\mathbf{x}$ has nearly the same expectation values in the center area, with a small perturbation in the marginal area due to padding. Therefore, $k_n \mathbf{J}_K$ as a kernel-wise approximation achieves a low approximate error for convolution. For any $\mathbb{E}[\mathbf{x}^\ell \mathbf{x}^{\ell T}]$, we can always find a decomposition that satisfies $e_{n,i}, k_n, c_m > 0$, for which we present the decomposition method in Appendix A.2. Substituting Eq. (5) into Eq. (4) yields the following equation.

$$\Delta \mathbf{W}_{m,:}^\ell \mathbb{E}[\mathbf{x}^\ell \mathbf{x}^{\ell T}] \Delta \mathbf{W}_{m,:}^{\ell T} \approx \sum_{n,i} e_{n,i} \Delta \mathbf{W}_{m,n,i}^{\ell}{}^2 + \sum_n k_n \Delta \mathbf{W}_{m,n,:}^\ell \mathbf{J}_K \Delta \mathbf{W}_{m,n,:}^{\ell T} + c_m \Delta \mathbf{W}_{m,:}^\ell \mathbf{J}_{NK} \Delta \mathbf{W}_{m,:}^{\ell T}. \tag{6}$$

### 3.3 DATA-FREE OPTIMIZATION

To achieve the data-free optimization objective, we omit the coefficients ($e_{n,i}$, $k_n$ and $c_m$) in Eq. (6), which leads to the approximate objective in Eq. (8) optimized by our fast SQuant framework. We present the omitting process and empirically verify that the approximation does almost not influence the performance in Appendix A.2 and Appendix A.3. It can be easily found that there are no training samples needed to minimize,

$$\underset{\Delta \mathbf{W}_{m,:}^\ell}{\arg\min} \quad \sum_{n,i} \Delta \mathbf{W}_{m,n,i}^{\ell}{}^2 + \sum_n \Delta \mathbf{W}_{m,n,:}^\ell \mathbf{J}_K \Delta \mathbf{W}_{m,n,:}^{\ell T} + \Delta \mathbf{W}_{m,:}^\ell \mathbf{J}_{NK} \Delta \mathbf{W}_{m,:}^{\ell T} \tag{7}$$

$$= \underset{\Delta \mathbf{W}_{m,:}^\ell}{\arg\min} \quad \sum_{n,i} \Delta \mathbf{W}_{m,n,i}^{\ell}{}^2 + \sum_n \left( \sum_i \Delta \mathbf{W}_{m,n,i}^\ell \right)^2 + \left( \sum_{n,i} \Delta \mathbf{W}_{m,n,i}^\ell \right)^2. \tag{8}$$

Next, we transform the overall objective Eq. (8) in the discrete space and explain how to compose and optimize the three approximated sub-items in order. Without loss of generality, we assume all weights have been scaled with the scale parameter $s_m$ for $\mathbf{W}_{m,:}^\ell$.

**Sub-item Analysis** For the element-wise item, i.e., the first item in Eq. (8), the problem is reduced to the following objective, which we call **SQuant-E**.

$$\Delta\widehat{\mathbf{W}}_{m,:}^\ell = \underset{\Delta\mathbf{W}_{m,:}^\ell}{\arg\min} \sum_{n,i} \Delta\mathbf{W}_{m,n,i}^{\ell\,2} = \underset{\Delta\mathbf{W}_{m,n,i}^\ell}{\arg\min} |\Delta\mathbf{W}_{m,n,i}^\ell| \Leftrightarrow \forall \Delta\widehat{\mathbf{W}}_{m,n,i}^\ell, \ |\Delta\widehat{\mathbf{W}}_{m,n,i}^\ell| \leq r_e = 0.5, \quad (9)$$

SQuant-E is essentially the rounding method when $r_e = 0.5$. Rounding does not introduce any approximate error and has $O(1)$ complexity for each weight element. However, as many previous works pointed out (Nagel et al., 2020), rounding-to-nearest is not optimal because it only considers the diagonal elements the matrix $\mathbb{E}[\mathbf{x}^\ell \mathbf{x}^{\ell^T}]$ while ignores the rest majority elements.

For kernel-wise item (the second item in Eq. (8)), we have the following objective called **SQuant-K**,

$$\Delta\widehat{\mathbf{W}}_{m,:}^\ell = \underset{\Delta\mathbf{W}_{m,:}^\ell}{\arg\min} \sum_n \left(\sum_i \Delta\mathbf{W}_{m,n,i}^\ell\right)^2 = \underset{\Delta\mathbf{W}_{m,n,:}^\ell}{\arg\min} |\sum_i \Delta\mathbf{W}_{m,n,i}^\ell| \Leftrightarrow \forall \Delta\widehat{\mathbf{W}}_{m,n,:}^\ell, \ |\sum_i \Delta\widehat{\mathbf{W}}_{m,n,i}^\ell| \leq r_k = 0.5, \quad (10)$$

where $|\sum_i \Delta\mathbf{W}_{m,n,i}^\ell|$ is the Absolute Sum of Error (**ASE**) of each kernel-wise weight matrix in the convolution and $r_k$ equals 0.5 because of the discrete quantization. In other words, **SQuant** is based on the insight of **S**um of (**S**igned) error instead of the accumulation of absolute (unsigned) error.

Similarly, for the output channel-wise item (the third item in Eq. (8)), we have **SQuant-C**,

$$\Delta\widehat{\mathbf{W}}_{m,:}^\ell = \underset{\Delta\mathbf{W}_{m,:}^\ell}{\arg\min} \left(\sum_{n,i} \Delta\mathbf{W}_{m,n,i}^\ell\right)^2 \Leftrightarrow \forall \Delta\widehat{\mathbf{W}}_{m,:}^\ell, \ |\sum_{n,i} \Delta\widehat{\mathbf{W}}_{m,n,i}^\ell| \leq r_c = 0.5. \quad (11)$$

**Relaxation** Obviously, $r_e = 0.5$ is against $r_k = 0.5$ because rounding ($r_e = 0.5$) only guarantees the upper-bound $\dot{r}_k = 0.5K$ for SQuant-K. Some elements need to relax the constraint $r_e$ to a larger number, such as 1.0, to satisfy $r_k = 0.5$. Similarly, SQuant-C also needs to relax $r_k = 1.0$.

**CASE Flipping** We adopt the flipping approach (Nagel et al., 2020) to minimize the ASE. Due to the discrete quantization, rounded elements can be flipped (from rounding up to rounding down and vice versa) with $\pm 1$ integer mutation. Formally, we need to work out a flipping set $\mathbf{f}_m$ to satisfy the overall objective Eq. (8) by composing these three sub-items in order (SQuant-E $\rightarrow$ SQuant-K $\rightarrow$ SQuant-C) with constraints relaxation. After optimization, the $\mathbf{f}_m$ will be

$$\forall (m,n,i) \notin \mathbf{f}_m, \ |\Delta\mathbf{W}_{m,n,i}^\ell| \leq 0.5; \quad \forall (m,n,j) \in \mathbf{f}_m, \ 0.5 \leq |\Delta\mathbf{W}_{m,n,j}^\ell| < 1.0, \quad (12)$$

where $\mathbf{f}_m$ is the index set of flipped elements for $m$-th output channel. Specifically, we need to flip $k = \lfloor \text{ASE} \rceil$ elements, whose perturbation has the same sign as $\sum_i \Delta\mathbf{W}_{m,n,i}^\ell$. We prove the equivalence for Eq. (10) and Eq. (11) by illustrating the transformation process to a discrete problem in Appendix B.1.

However, any $k$ elements can satisfy Eq. (10) leading to large search space. Fortunately, based on Eq. (9), SQuant-K can select specific $k$ elements with the top-$k$ largest perturbation because they will have the smallest perturbation after flipping under the constraint of SQuant-E. Therefore, we adopt the Constrained ASE (**CASE**) to optimize the SQuant-E&K composition via the top-$k$ perturbation algorithm, which is the only solution for minimizing the CASE proven in Appendix B.2. Obviously, SQuant-E&K&C needs to "flip" the "SQuanted" kernel after SQuant-E&K. Notice that we can only flip one element for a kernel to satisfy the constraint $r_k = 1.0$.

The following section will introduce an efficient SQuant algorithm with a linear computation complexity for CASE flipping.

### 3.4 On-the-Fly SQuant

**Progressive Algorithm** We design a progressive algorithm illustrated in algorithm 1 to meet our stated optimization objective, i.e., minimizing the CASE of weight perturbation. The critical insight of the progressive algorithm is to gradually calibrate the deviation from the optimal global solution introduced by the fine-grained diagonal sub-item. To calibrate the SQuant-E, SQuant-K flips certain rounded elements. After the SQuant-K calibration, SQuant-C then further flips SQuanted kernels.

We start by rounding the weight and updating its perturbation to satisfy $r_e = 0.5$ (Line 4-5). Then we run the SQuant-K (Line 6) to flip specific elements under $r_e = 1.0$, satisfy $r_k = 0.5$, and update kernel perturbation (Line 7). The follow-up SQuant-C (Line 8) further flips specific kernels under $r_k = 1.0$ and satisfy $r_c = 0.5$. Finally, we derive the quantized weights (Line 9).

---

**Algorithm 1:** Progressive SQuant Algorithm.

---

**Input:** Weight tensor $\mathbf{W}$ of layer $\ell$, scale factor $\mathbf{s}$ of layer $\ell$.
**Output:** Quantized weight tensor $\mathbf{C}$ of layer $\ell$.

1 **foreach** $m \in [1, .., M]$ **do** // SQuant-C: SQuant $M$ output channels.
2     **foreach** $n \in [1, .., N]$ **do** // SQuant-K: SQuant $N$ kernels.
3        **foreach** $i \in [1, ..., K]$ **do** // SQuant-E: Round $K$ elements.
4           $\mathbf{E}_{m,n,i} = \lfloor \mathbf{W}_{m,n,i}/\mathbf{s}_m \rceil$ // Scale and SQuant-E (Rounding).
5           $\Delta\mathbf{E}_{m,n,i} = \mathbf{E}_{m,n,i} - \mathbf{W}_{m,n,i}$ // Element perturbation.
6        $\mathbf{K}_{m,n,:} = \text{SQuantFlip}(\mathbf{E}_{m,n,:}, \Delta\mathbf{E}_{m,n,:})$// SQuant-K.
7        $\Delta\mathbf{K}_{m,n,:} = \text{UpdatePerturbation}(\Delta\mathbf{E}_{m,n,:})$ // Kernel perturbation.
8     $\mathbf{C}_{m,:} = \text{SQuantFlip}(\mathbf{K}_{m,:}, \Delta\mathbf{K}_{m,:}) \cdot \mathbf{s}_m$// SQuant-C.
9 **return C**

---

**Flip Algorithm** SQuant-K and SQuant-C can utilize the same flip function. The goal of the flip algorithm is to find a proper element set $\mathbf{f}$ to flip and minimize the CASE depicted in algorithm 2. First, we need to compute the accumulated perturbation ($e$) (Line 2). We select weights with positive perturbation to decrease the positive $e$ and vice versa for negative $e$. Therefore, we set 0 for the elements with a different sign (Line 3) to disable them. Obviously, we need only $k = \lfloor |e| \rceil$ elements and reduce $|e| < r_k = 0.5$ (Line 4). Finally, we flip $k$ weights with the largest $|\mathbf{p}|$ (Line 5-6). For now, we have SQuanted the kernel and tuned kernel CASE to $|e| \le r_k = 0.5$. Specifically, for FC and Conv with a kernel size of $(1, 1)$, we can skip the SQuant-K. As mentioned in Section 3.3, SQuant-C flips only one element in each kernel. Therefore, we update the kernel perturbation (Line 7 of algorithm 1) for SQuant-C to flip kernel illustrated in Appendix B.3. As a result, SQuant successfully identifies the optimum combination $\mathbf{f}$ under a low computation complexity, which we analyze in Appendix B.4.

---

**Algorithm 2:** SQuant Flip Algorithm.

---

**Input:** Rounded/SQuanted Weight $\mathbf{w}$;
        Weight perturbation $\mathbf{p}$.
**Output:** Updated Quantized Weight $\mathbf{w}$.

1 **def** SQuantFlip($\mathbf{w}$, $\mathbf{p}$):
2     $e = \sum_i \mathbf{p}_i$// Accumulated perturbation.
3     $\mathbf{p}[e \cdot \mathbf{p} < 0] = 0$ // Disable Elements/kernels with different sign from $e$.
4     $k = \lfloor |e| \rceil$ // Flip $k$ elements/kernels based on the CASE.
5     $\mathbf{f} = \text{TopK}(|\mathbf{p}|, k).\text{indices}$// Indices of $k$ largest perturbation.
6     $\mathbf{w}[\mathbf{f}] = \text{Flip}(\mathbf{w}[\mathbf{f}])$// Flip $k$ elements/kernels with same sign as $e$.
7     **return w**

---

**On-the-Fly Framework** From the overall perspective of the optimization, SQuant-K has $MN$ sub-problems, while SQuant-C has $M$ sub-problems. Because of the independence of sub-problems, SQuant is friendly for DNN accelerators, e.g., GPU, allowing each sub-problem to be accelerated in parallel. Without the requirement of back-propagation nor fine-tuning, SQuant can run on inference-only devices with constrained computation and memory resources on the fly. That provides new opportunities for optimizing weight quantization. In the next section, we demonstrate the impressive efficiency and accuracy of SQuant.

## 4 EXPERIMENTS

For demonstrating the strength of SQuant, we evaluate the SQuant as well as four SOTA methods, DFQ (Nagel et al., 2019), ZeroQ (Cai et al., 2020), DSG (Zhang et al., 2021; Qin et al., 2021), and

| Arch | Method | No BP | No FT | W-bit | A-bit | Top-1 |
|---|---|---|---|---|---|---|
| | Baseline | – | – | 32 | 32 | 71.47 |
| ResNet18 | DFQ | ✓ | ✓ | 4 | 4 | 0.10 |
| | ZeroQ | ✗ | ✓ | 4 | 4 | 19.09 |
| | DSG | ✗ | ✓ | 4 | 4 | 34.53 |
| | GDFQ | ✗ | ✗ | 4 | 4 | 60.60 |
| | SQuant | ✓ | ✓ | 4 | 4 | **66.14** |
| | DFQ | ✓ | ✓ | 6 | 6 | 67.30 |
| | ZeroQ | ✗ | ✓ | 6 | 6 | 69.84 |
| | DSG | ✗ | ✓ | 6 | 6 | 70.46 |
| | GDFQ | ✗ | ✗ | 6 | 6 | 70.13 |
| | SQuant | ✓ | ✓ | 6 | 6 | **70.74** |
| | DFQ | ✓ | ✓ | 8 | 8 | 69.70 |
| | ZeroQ | ✗ | ✓ | 8 | 8 | 71.43 |
| | GDFQ | ✗ | ✗ | 8 | 8 | 70.68 |
| | SQuant | ✓ | ✓ | 8 | 8 | **71.47** |
| | Baseline | – | – | 32 | 32 | 77.74 |
| ResNet50 | ZeroQ | ✗ | ✓ | 4 | 4 | 7.75 |
| | DSG | ✗ | ✓ | 4 | 4 | 23.10 |
| | GDFQ | ✗ | ✗ | 4 | 4 | 55.65 |
| | SQuant | ✓ | ✓ | 4 | 4 | **70.80** |
| | ZeroQ | ✗ | ✓ | 6 | 6 | 72.93 |
| | DSG | ✗ | ✓ | 6 | 6 | 76.07 |
| | GDFQ | ✗ | ✗ | 6 | 6 | 76.59 |
| | SQuant | ✓ | ✓ | 6 | 6 | **77.05** |
| | ZeroQ | ✗ | ✓ | 8 | 8 | 77.65 |
| | DSG | ✗ | ✓ | 8 | 8 | 77.68 |
| | GDFQ | ✗ | ✗ | 8 | 8 | 77.51 |
| | SQuant | ✓ | ✓ | 8 | 8 | **77.71** |

Table 1: Results of data-free methods with ResNet18 and ResNet50. "No BP" means that no back-propagation algorithm is used to generate data, "No FT" means no fine-tuning (retraining) for weight quantization.

| Arch | Method | No BP | No FT | W-bit | A-bit | Top-1 |
|---|---|---|---|---|---|---|
| | Baseline | – | – | 32 | 32 | 78.81 |
| Inception V3 | ZeroQ | ✗ | ✓ | 4 | 4 | 18.20 |
| | GDFQ | ✗ | ✗ | 4 | 4 | 70.39 |
| | SQuant | ✓ | ✓ | 4 | 4 | **73.26** |
| | ZeroQ | ✗ | ✓ | 6 | 6 | 74.94 |
| | GDFQ | ✗ | ✗ | 6 | 6 | 77.20 |
| | SQuant | ✓ | ✓ | 6 | 6 | **78.30** |
| | ZeroQ | ✗ | ✓ | 8 | 8 | 78.78 |
| | GDFQ | ✗ | ✗ | 8 | 8 | 78.62 |
| | SQuant | ✓ | ✓ | 8 | 8 | **78.79** |
| | Baseline | – | – | 32 | 32 | 69.38 |
| Squeeze Next | ZeroQ | ✗ | ✓ | 4 | 4 | 0.09 |
| | GDFQ | ✗ | ✗ | 4 | 4 | 28.93 |
| | SQuant | ✓ | ✓ | 4 | 4 | **43.45** |
| | ZeroQ | ✗ | ✓ | 6 | 6 | 16.54 |
| | GDFQ | ✗ | ✗ | 6 | 6 | 65.46 |
| | SQuant | ✓ | ✓ | 6 | 6 | **67.34** |
| | ZeroQ | ✗ | ✓ | 8 | 8 | 68.18 |
| | GDFQ | ✗ | ✗ | 8 | 8 | 68.22 |
| | SQuant | ✓ | ✓ | 8 | 8 | **69.22** |
| | Baseline | – | – | 32 | 32 | 65.07 |
| Shuffle Net | ZeroQ | ✗ | ✓ | 6 | 6 | 35.21 |
| | GDFQ | ✗ | ✗ | 6 | 6 | 60.12 |
| | SQuant | ✓ | ✓ | 6 | 6 | **60.25** |
| | ZeroQ | ✗ | ✓ | 8 | 8 | 64.34 |
| | GDFQ | ✗ | ✗ | 8 | 8 | 64.03 |
| | SQuant | ✓ | ✓ | 8 | 8 | **64.68** |

Table 2: Results of data-free methods with InceptionV3, SqueezeNext, and ShuffleNet.

GDFQ (Xu et al., 2020), with 5 different CNN models including ResNet-18 & 50 (He et al., 2016), Inception V3 (Szegedy et al., 2016), SqueezeNext (Gholami et al., 2018) and ShuffleNet (Zhang et al., 2018) on the golden standard dataset ImageNet (Krizhevsky et al., 2012).

In our experiments, SQuant is dedicated to weight quantization, including setting quantization range and selecting the grid point with per-channel quantization, which is friendly for hardware accelerators. With the BN-based approach, we adopt a simple rounding method and a wide quantization range for activation suggested by DFQ (Nagel et al., 2019) without breaking the data-free premise. We clip activation tensors in a layerwise manner (per-tensor). We utilize a uniform distribution as the initialization for the activation quantization range. All DFQ algorithms are implemented with PyTorch (Paszke et al., 2019) and evaluated on Nvidia GPU A100-40GB. Unless otherwise stated, we employ both weight and activation quantization in all experiments. Also, uniform quantization grids are used in all experiments, and hyper-parameters, e.g., $r_e = r_k = 1.0$ and $r_c = 0.5$, for all SQuant experiments are the same.

## 4.1 COMPARISON TO SOTA METHODS

Table 1 and Table 2 show the results on the ImageNet datasets for various bit-width choices, comparing our SQuant against other data-free methods. Among these methods, ZeroQ, DSG, and GDFQ are data-generative approaches with back-propagation. The former two are PTQ methods, while the last is a QAT method, which retrains the network with the synthetic data. DFQ is the only true data-free method with weight equalization and bias correction.

Experiments show that SQuant significantly outperforms all other SOTA DFQ methods, even with synthetic dataset calibrating their networks. The 8-bit quantization preserves better network accuracy

| Arch | ResNet18 | ResNet50 | InceptionV3 | SqueezeNext | ShuffleNet |
|---|---|---|---|---|---|
| Layers | 21 | 54 | 95 | 112 | 50 |
| SQuant Time (**ms**) | 84 | 188 | 298 | 272 | 121 |
| ZeroQ Time (**s**) | 38 | 92 | 136 | 109 | 38 |
| GDFQ Time (**hour**) | 1.7 | 3.1 | 5.7 | 4.8 | 1.9 |

Table 3: SQuant, ZeroQ and GDFQ 4-bit quantization time on GPU A100

than the lower-bit quantization does because of higher precision. The benefit of SQuant becomes more prominent as the bit-width decreases. SQuant outperforms the PTQ methods, i.e., DFQ, ZeroQ, and DSG, more than 30% on all models with 4-bit quantization. It is noteworthy that SQuant surpasses GDFQ in all cases and even surpasses more than 15% in ResNet50 under 4-bit quantization, although GDFQ is a quantization-aware training method.

Table 1 and Table 2 also show that GDFQ significantly outperforms ZeroQ and DSG under lower-bit settings (e.g., 4-bit). Since we use the same activation quantization method for evaluating these methods, the results indicate that the weight quantization plays a critical role in the overall model quantization. However, GDFQ requires fine-tuning (FT) with back-propagation (BP). In contrast, SQuant adopts a direct optimization objective of weight perturbation, which does not require fine-tuning nor BP, and still outperforms GDFQ in the 4-bit setting. These results clearly illustrate the advantages of SQuant, a CASE-based optimization framework, which is to minimize the CASE of weight perturbation.

## 4.2 SQUANT EFFICIENCY

The trade-off between efficiency and accuracy is challenging for previous DFQ methods. Before SQuant, DFQ is the fastest one since it does not require back-propagation and fine-tuning, but it performs poorly, especially in low-bit cases. GDFQ performs relatively well but takes hours to complete 400 epochs that produce synthetic data from weights and fine-tune the network. SQuant employs the direct optimization objective, minimizing the CASE of weight perturbation, pushes the quantization procedure to a sub-second level. Table 3 shows the 4-bit quantization time of the five models using SQuant, ZeroQ, and GDFQ. The efficient algorithm design also contributes to the surprising results. Note that the SQuant results in Table 3 are the sum of all layer quantization time, and it will be faster if we quantize layers in parallel. A single layer takes SQuant just 3 milliseconds on average because SQuant does not involve complex algorithms, such as back-propagation and fine-tuning. That means we can implement the SQuant algorithm on inference-only devices such as smartphones and IoT devices and quantize the network on the fly.

## 4.3 ABLATION STUDY

**SQuant Granularity** We decouple the effect of SQuant-K and SQuant-C, which have different granularities to optimize CASE. As shown in Table 4, their accuracies both outperform SQuant-E

| Method | W-bit | A-bit | Top-1 |
|---|---|---|---|
| Baseline | 32 | 32 | 71.47 |
| SQuant-E | 3 | 32 | 2.05 |
| SQuant-E&C | 3 | 32 | 40.87 |
| SQuant-E&K | 3 | 32 | 52.07 |
| SQuant-E&K&C | 3 | 32 | **60.78** |
| SQuant-E | 4 | 32 | 48.15 |
| SQuant-E&C | 4 | 32 | 67.14 |
| SQuant-E&K | 4 | 32 | 68.07 |
| SQuant-E&K&C | 4 | 32 | **69.75** |

Table 4: SQuant ablation results with ResNet18.

| Method | No BP | No SD | W-bit | A-bit | Top-1 |
|---|---|---|---|---|---|
| Baseline | – | – | 32 | 32 | 71.47 |
| ZeroQ + AdaRound | ✗ | ✗ | 3 | 32 | 49.86 |
| DSG + AdaRound | ✗ | ✗ | 3 | 32 | 56.09 |
| SQuant | ✓ | ✓ | 3 | 32 | **60.78** |
| ZeroQ + AdaRound | ✗ | ✗ | 4 | 32 | 63.86 |
| DSG + AdaRound | ✗ | ✗ | 4 | 32 | 66.87 |
| SQuant | ✓ | ✓ | 4 | 32 | **69.75** |
| ZeroQ + AdaRound | ✗ | ✗ | 5 | 32 | 68.39 |
| DSG + AdaRound | ✗ | ✗ | 5 | 32 | 68.97 |
| SQuant | ✓ | ✓ | 5 | 32 | **71.19** |

Table 5: ResNet18 results of SQuant, ZeroQ and DSG with AdaRound. "No SD" means no synthetic data.

(i.e., rounding), and combining them leads to higher accuracy for ResNet18. SQuant-E&C has a lower accuracy than SQuant-E&K because SQuant-C has a more significant approximation error than SQuant-K. On the other hand, SQuant-E alone is not optimal because it uses a smaller granularity and ignores a large amount of Hessian information as we analyze in Section 3. This ablation study shows that SQuant-E&K&C achieves the best accuracy by exploiting the most Hessian information (`H−C`), and SQuant-E&K also achieves a higher accuracy with `H−K` than SQuant-E with `H−E`.

**Comparison to Data-free AdaRound** AdaRound (Nagel et al., 2020) is a novel data-driven PTQ method, which also utilizes the Hessian-based approach to round-up or round-down the weights with an approximation assumption. Under the data-free premise, we augment ZeroQ and DSG with the AdaRound by feeding their generated synthetic data to AdaRound. Results in Table 5 show that SQuant has better accuracy than data-free AdaRound because SQuant directly optimizes the CASE objective instead of the MSE of the output activation adopted by AdaRound. It is hard for DSG+AdaRound to find the optimal solution with an excessively long optimization path and gradient-based approaches. Even though AdaRound tries a shorter way to fine-tune the weights in the layer-wise fashion, SQuant still outperforms AdaRound in quantization time and accuracy.

## 5 RELATED WORK

Compression is a promising method to reduce the DNN model's memory and computation cost. Pruning (Han et al., 2015b;a) is one of the effective approaches to exploit the inherent redundancy of DNN. However, pruning will cause sparse irregular memory accesses. Therefore, pruning needs software (Gale et al., 2020; Guan et al., 2020; Qiu et al., 2019; Guo et al., 2020a; Guan et al., 2021; Fedus et al., 2021) and hardware (Gondimalla et al., 2019; Guo et al., 2020b; Zhang et al., 2020; Wang et al., 2021) optimization to accelerate.

Quantization is more practical because it can be supported directly by existing accelerators. Quantization-aware training (QAT) (Gupta et al., 2015; Jacob et al., 2018; Wang et al., 2019; Zhuang et al., 2021) is one of the most promising techniques to retrain networks and mitigate the accuracy drop introduced by quantization. However, the training procedure is time-consuming and costly. Therefore, post-training quantization (PTQ) (Banner et al., 2018; Choukroun et al., 2019; Zhao et al., 2019; Nagel et al., 2020) has earned lots of attention due to the absence of any fine-tuning or retraining process, at the expense of accuracy.

Recently, several methods for CNN quantization without the original training datasets have been proposed. These methods are known as data-free quantization (DFQ), including PTQ (Nagel et al., 2019; Cai et al., 2020; Zhang et al., 2021) and QAT (Xu et al., 2020; Liu et al., 2021; Qin et al., 2021; Choi et al., 2020). DFQ (Nagel et al., 2019) and ACIQ (Nagel et al., 2019) rely on weight equalization or bias correction without requiring synthetic data. Other works synthesize the data to calibrate or fine-tune the network based on the batch normalization statistics (Cai et al., 2020) or adversarial knowledge distillation techniques (Liu et al., 2021; Choi et al., 2020).

## 6 CONCLUSION

This paper approximates and composes the original Hessian optimization objective into the CASE of weight perturbation with a data-free premise. Surprisingly, CASE only involves the weight perturbation and requires no knowledge of any datasets or network architecture. Based on that, we proposed the on-the-fly SQuant framework. We used a progressive algorithm to minimize CASE directly and significantly improve accuracy than other DFQ methods. SQuant considerably reduces optimization complexity and accelerates the data-free quantization procedure, which previously requires back-propagation with massive computation and memory resources consumption seen in other works. In summary, SQuant outperforms other data-free quantization approaches in terms of accuracy and pushes the quantization processing time to a sub-second level.

### ACKNOWLEDGMENTS

We would like to thank the anonymous reviewers for their constructive feedback. This work was supported by the National Key R&D Program of China under Grant 2021ZD0110104, and the National Natural Science Foundation of China (NSFC) grant (U21B2017, 62072297, and 61832006).

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

## A   APPROXIMATION AND DECOMPOSITION

### A.1   APPROXIMATED HESSIAN MATRIX FOR DATA-FREE QUANTIZATION

The quantization loss function for the entire network is

$$\mathcal{L}(\Delta \mathbf{W}) = \Delta \mathbf{W} \, \mathbb{E}[\mathbf{H}] \Delta \mathbf{W}^T. \tag{13}$$

Consider a convolution layer defined as

$$\mathbf{Y}_{m,h,w} = \sum_{n,i,j} \mathbf{W}_{m,n,i,j} \mathbf{X}_{n,h-i,w-j}. \tag{14}$$

Here, $\mathbf{Y}$ has three dimensions, output channel, output feature map height, and output feature map width, i.e., $(M \times OH \times OW)$ indexing by $(m,h,w)$, $\mathbf{W}$ has four dimensions, output channel, input channel, kernel height, kernel width, i.e., $(M \times N \times KH \times KW)$ indexing by $(m,n,i,j)$, and $\mathbf{X}$ has three dimensions, input channel, input feature map height, and input feature map width, i.e., $(N \times IH \times IW)$ indexing by $(n,h-i,w-j)$. Ignoring the interaction between layers and output

channels following Nagel et al. (2020), for a specific convolution layer $l$ and output channel $m$, the elements of corresponding output channel-wise Hessian $\mathbf{H}^{\mathbf{W}_m^\ell}$ is

$$\mathbf{H}^{\mathbf{W}_m^\ell}_{n,i,j,n',i',j'} = \frac{\partial^2 \mathscr{L}}{\partial \mathbf{W}_{m,n,i,j} \partial \mathbf{W}_{m,n',i',j'}} \tag{15}$$

$$= \frac{\partial}{\partial \mathbf{W}_{m,n,i,j}} \sum_{h,w} \frac{\partial \mathscr{L}}{\partial \mathbf{Y}_{m,h,w}} \frac{\partial \mathbf{Y}_{m,h,w}}{\partial \mathbf{W}_{m,n,i,j}} \tag{16}$$

$$= \frac{\partial}{\partial \mathbf{W}_{m,n,i,j}} \sum_{h,w} \frac{\partial \mathscr{L}}{\partial \mathbf{Y}_{m,h,w}} \mathbf{X}_{n,h-i,w-j} \tag{17}$$

$$= \sum_{h,w} \left( \frac{\partial}{\partial \mathbf{Y}_{m,h,w}} \frac{\partial \mathscr{L}}{\partial \mathbf{Y}_{m,h',w'}} \right) \mathbf{X}_{n,h-i,w-j} \tag{18}$$

$$= \sum_{h,w} \left( \frac{\partial}{\partial \mathbf{Y}_{m,h,w}} \sum_{h',w'} \frac{\partial \mathscr{L}}{\partial \mathbf{Y}_{m,h',w'}} \mathbf{X}_{n',h'-i',w'-j'} \right) \mathbf{X}_{n,h-i,w-j} \tag{19}$$

$$= \sum_{h,w} \sum_{h',w'} \frac{\partial^2 \mathscr{L}}{\partial \mathbf{Y}_{m,h,w} \partial \mathbf{Y}_{m,h',w'}} \mathbf{X}_{n,h-i,w-j} \mathbf{X}_{n',h'-i',w'-j'} \tag{20}$$

Assuming $\nabla^2_{\mathbf{y}^\ell} \mathscr{L}$ is a diagonal matrix yields Eq. (30) in (Nagel et al., 2020)

$$\mathbf{H}^{\mathbf{W}_m^\ell}_{n,i,j,n',i',j'} \approx \sum_{h,w} \frac{\partial^2 \mathscr{L}}{\partial \mathbf{Y}^2_{m,h,w}} \mathbf{X}_{n,h-i,w-j} \mathbf{X}_{n,h-i',w-j'}. \tag{21}$$

To make Eq. (13) get irrelevant to training samples, we assume that input feature maps auto-correlate with each other in a similar way, resulting in

$$\mathbb{E}[\mathbf{H}^{\mathbf{W}_m^\ell}_{n,i,j,n',i',j'}] = \mathbb{E}[\sum_{h,w} \frac{\partial^2 \mathscr{L}}{\partial \mathbf{Y}^2_{m,h,w}} \mathbf{X}_{n,h-i,w-j} \mathbf{X}_{n',h-i',w-j'}] \approx c_m \tag{22}$$

for all $n, i, j, n', i'$ and $j'$, where $c_m$ is a constant. It should be noted that Eq. (22) is a strong assumption. For more accurate approximation, we further look into each input channel (i.e., $n = n'$, $i \neq i'$, and $j \neq j'$), and find

$$\mathbb{E}[\mathbf{H}^{\mathbf{W}_{m,n}^\ell}_{i,j,i',j'}] = \mathbb{E}[\sum_{h,w} \frac{\partial^2 \mathscr{L}}{\partial \mathbf{Y}^2_{m,h,w}} \mathbf{X}_{n,h-i,w-j} \mathbf{X}_{n,h-i',w-j'}] \approx k_{m,n}, \tag{23}$$

for all $i$, $j$, $i'$, and $j'$, where $k_{m,n}$ is a constant. It is generally true because the kernel size is usually much smaller than the size of feature maps, so the shift introduced by different $i$, $j$, $i'$, and $j'$ is a small perturbation of $\mathbb{E}[\mathbf{H}^{\mathbf{W}_{m,n}^\ell}_{i,j,i',j'}]$ compared with the summation over the entire feature map. Finally, we focus on each diagonal element of Hessian matrix (i.e., $n = n'$, $i = i'$, and $j = j'$) and denote

$$\mathbb{E}[\mathbf{H}^{\mathbf{W}_{m,n,i,j}^\ell}] = \mathbb{E}[\sum_{h,w} \frac{\partial^2 \mathscr{L}}{\partial \mathbf{Y}^2_{m,h,w}} \mathbf{X}^2_{n,h-i,w-j}] = e_{m,n,i,j}, \tag{24}$$

where $e_{m,n,i,j}$ is a constant. Please note that the output channel-wise expected Hessian matrix $\mathbb{E}[\mathbf{H}^{\mathbf{W}_m^\ell}_{n,i,j,n',i',j'}]$ is a principle submatrix of $\mathbb{E}[\mathbf{H}]$, so it must positive semi-define. Therefore, we set $e_{m,n,i,j} > k_{m,n} > c_m > 0$ to ensure the approximation to $\mathbb{E}[\mathbf{H}^{\mathbf{W}_m^\ell}_{n,i,j,n',i',j'}]$ is also positive semi-define and nontrivial. Considering Eq. (22), Eq. (23), and Eq. (24) at the same time, we can get the approximation to expected Hessian shown in Eq. (5). Extending the discussion to fully connected layer is straightforward thus omitted here.

## A.2 DECOMPOSITION

In this section, we present the decomposition method for $\mathbf{H}^{\mathbf{W}_m^\ell} = l_m \mathbb{E}[\mathbf{x}^\ell \mathbf{x}^{\ell T}]$ illustrated in the algorithm 3. First, we construct three matrices with the shape of $(NK, NK)$, $\mathbf{C}' = \mathbf{J}_{NK}$,

$$\mathbf{K}' = \begin{bmatrix} \mathbf{J}_K & & \\ & \ddots & \\ & & \mathbf{J}_K \end{bmatrix}, \text{ and } \mathbf{E}' = \begin{bmatrix} 1 & & \\ & \ddots & \\ & & 1 \end{bmatrix}.$$

Here, $\mathbf{J}_{NK}$ is an all-one matrix with dimension $NK = N \times K$ and $\mathbf{J}_K$ represents an all-one matrix with dimension $K$. The $n$-th diagonal block corresponds to $n$-th kernel in convolution and has the same constant $k_n$. $\mathbf{E}$ is a diagonal matrix whose diagonal elements are 1.

---

**Algorithm 3:** $\mathbb{E}[\mathbf{x}^\ell \mathbf{x}^{\ell T}]$ Decomposition.

---

**Input:** $\mathbb{E}[\mathbf{x}^\ell \mathbf{x}^{\ell T}]$, $\mathbf{H}$;
    Channel-wise matrix, $\mathbf{C}'$.
    Kernel-wise matrix, $\mathbf{K}'$.
    Element-wise matrix, $\mathbf{E}'$.
**Output:** Matrix $\mathbf{E}, \mathbf{K}$, and $\mathbf{C}$.

1   $\mathbf{H}' = |\mathbf{H}|$
2   $c_m = (1 - \varepsilon) \cdot \min(\mathbf{H}')$   // Output channel-wise.
3   $\mathbf{C} = c_m \mathbf{C}'$
4   **foreach** $n \in [1, ..., N]$ **do** // Kernel-wise.
5      $k_n = (1 - \varepsilon_n') \cdot \min(\mathbf{H}'_{n:n+K, \, n:n+K} - c_m)$
6      $\mathbf{K}_{n,:} = k_n \mathbf{K}'_{n,:}$
7      **foreach** $i \in [1, ..., K]$ **do** // Element-wise.
8         $e_{n,i} = \mathbf{H}'_{n \times K+i, \, n \times K+i} - c_m - k_n$
9         $\mathbf{E}_{n,i} = e_{n,i} \mathbf{E}'_{n,i}$

10   **return** $\mathbf{E}, \mathbf{K}, \mathbf{C}$

---

In algorithm 3, $0 < \varepsilon, \varepsilon' < 1$, and we can get the matrices $\mathbf{E}, \mathbf{K}$, and $\mathbf{C}$. Evidently, algorithm 3 can make $c_m > 0$, $k_n > 0$, and $e_{n,i} > 0$ for any $\mathbb{E}[\mathbf{x}^\ell \mathbf{x}^{\ell T}] \approx \mathbf{E} + \mathbf{K} + \mathbf{C}$.

## A.3 APPROXIMATION ERROR ANALYSIS

To achieve the data-free optimization objective, we omit the coefficients ($e_{n,i}$, $k_n$ and $c_m$) in Eq. (6), which leads to the approximate objective in Eq. (8) optimized by our fast SQuant framework. We approximate Eq. (6) to Eq. (8) to enable fast data-free quantization. The approximation error is insignificant as our comprehensive results have shown the high accuracy of the final quantized model in Table 1 and Table 2 of the manuscript. The intuition behind the approximation is that we use an iterative process which progressively reduces each term of Eq. (6). Because each term's coefficient ($e_{n,i}$, $k_n$, and $c_m$) is positive, the reduction of each term would generally lead to the reduction of the precise objective in Eq. (6). In this section, we provide an empirical analysis of the approximation error between Eq. (6) andEq. (8).

In this empirical experiment, we use the real dataset to generate the precise coefficients of $e_{n,i}$, $k_n$, and $c_m$ in Eq. (6). To quantify the approximation error in our SQuant framework, we evaluate a metric called approximation precision and show that we achieve a nearly 95% approximation precision.

Since SQuant uses the flipping-based iterative optimization framework to minimize Eq. (8), we define an element as correctly flipped if its flipping leads to the decrease of the precise objective Eq. (6) and approximate objective Eq. (8). The approximation precision (AP) is the ratio of the correct element based on data-free Eq. (8) compared to data-driven Eq. (6), i.e.,

$$\text{AP} = \frac{\text{Number of correct elements}}{\text{Number of flipped elements}}.$$

| Layers | Squant-E&K | | | Squant-E&K&C | | |
|--------|---------|---------|---------|---------|---------|---------|
| | Flipped | Correct | AP | Flipped | Correct | AP |
| 1 | 2346 | 2346 | 100.00 % | 123 | 123 | 100.0 % |
| 2 | 2683 | 2594 | 96.68 % | 97 | 97 | 100.0 % |
| 3 | 2662 | 2653 | 99.66 % | 100 | 100 | 100.0 % |
| 4 | 2698 | 2630 | 97.48 % | 107 | 107 | 100.0 % |
| 5 | - | - | - | 230 | 220 | 95.7 % |
| 6 | 5349 | 5339 | 99.81 % | 223 | 223 | 100.0 % |
| 7 | 10782 | 10676 | 99.02 % | 332 | 332 | 100.0 % |
| 8 | 10777 | 10633 | 98.66 % | 342 | 342 | 100.0 % |
| 9 | 10655 | 10424 | 97.83 % | 348 | 348 | 100.0 % |
| 10 | - | - | - | 649 | 619 | 95.4 % |
| 11 | 21371 | 21173 | 99.07 % | 663 | 663 | 100.0 % |
| 12 | 43116 | 41561 | 96.39 % | 906 | 906 | 100.0 % |
| 13 | 42976 | 41402 | 96.34 % | 932 | 932 | 100.0 % |
| 14 | 43321 | 41315 | 95.37 % | 918 | 918 | 100.0 % |
| 15 | - | - | - | 1988 | 1639 | 82.4 % |
| 16 | 86010 | 83070 | 96.58 % | 1846 | 1846 | 100.0 % |
| 17 | 171344 | 161358 | 94.17 % | 2629 | 2629 | 100.0 % |
| 18 | 172071 | 158066 | 91.86 % | 2602 | 2602 | 100.0 % |
| 19 | 172623 | 154536 | 89.52 % | 2504 | 2504 | 100.0 % |
| Total | 800784 | 749776 | 93.6 % | 17539 | 17150 | 97.8 % |
| Acc. | | 68.07 | | | 69.75 | |

Table 6: ResNet18 results under 4-bit weight-only quantization. "Flipped" is the number of the Flipped elements after SQuant optimization. "Correct" is the number of elements has the same optimization direction as the precise objective. AP is the approximation precision.

We perform the above approximation error analysis on ResNet18 with ImageNet under 4-bit weight-only quantization. We evaluate the SQuanted weight on the inference datasets. We compute the coefficients $e_{n,i}$, $k_n$, and $c_m$ with 1000 samples. Table 6 shows the results, which clearly show that SQuant-E&K&C achieves nearly 100% approximation precision. In other words, nearly all flipped elements can indeed reduce the precise objective in Eq. (6). Based on this empirical study, we show that the approximation from Eq. (6) to Eq. (8) is effective for our data-free quantization.

# B SQUANT ALGORITHM

## B.1 DISCRETE OPTIMIZATION PROBLEM

We introduce the transformation of the discrete optimization problem. We know that the quantization is to round the scaled elements to the integer grid. Each quantization step has two rounding directions, rounding up and down, with a step size of 1. We have the quantization example within $[0, 1]$ shown in Fig. 2.

Clearly, the element 0.7 (0.4) can be rounded up (down) to 1.0 (0.0) with $+0.3$ ($-0.4$) orignal perturbation and flipped to 0.0 (1.0) with $-0.7$ ($+0.6$) flipped perturbation with $-1$ ($+1$) mutation. Each flipping operation leads to a $\pm 1$ integer mutation and increases the perturbation to $[0.5, 1.0]$. We prove that we can always find $k = \lfloor |\sum_i \Delta \mathbf{W}_{m,n,i}^\ell| \rceil$ elements to flip and reduce $|\sum_i \Delta \widehat{\mathbf{W}}_{m,n,i}^\ell| \le 0.5$.

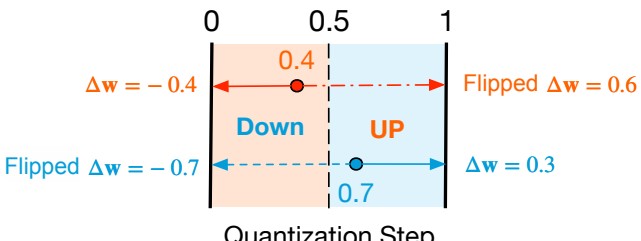

Figure 2: The flipping approach.

*Proof.* Assume $n$-th kernel has $K$ elements, $a$ rounded up elements with index set $\mathbf{f}_a$ have positive perturbation, $b$ rounded down elements with index set $\mathbf{f}_b$ have negative perturbation, and $K = a + b$. Then, we have

$$|\sum_i \Delta \mathbf{W}^\ell_{m,n,i}| = \left|\sum_t \Delta \mathbf{W}^\ell_{m,n,t} - \sum_j |\Delta \mathbf{W}^\ell_{m,n,j}|\right|, \quad t \in \mathbf{f}_a, \quad j \in \mathbf{f}_b \tag{25}$$

$$\leq \max(\sum_t \Delta \mathbf{W}^\ell_{m,n,t}, \quad \sum_j |\Delta \mathbf{W}^\ell_{m,n,j}|), \quad t \in \mathbf{f}_a, \quad j \in \mathbf{f}_b \tag{26}$$

Without loss of generality, let $\sum_t \Delta \mathbf{W}^\ell_{m,n,t} > \sum_j |\Delta \mathbf{W}^\ell_{m,n,j}|$,

$$\max(\sum_t \Delta \mathbf{W}^\ell_{m,n,t}, \quad \sum_j |\Delta \mathbf{W}^\ell_{m,n,j}|) = \sum_t \Delta \mathbf{W}^\ell_{m,n,t} \quad t \in \mathbf{f}_a, \quad j \in \mathbf{f}_b \tag{27}$$

$$\leq 0.5 \cdot a. \tag{28}$$

Therefore, we can always find $k = \lfloor|\sum_i \Delta \mathbf{W}^\ell_{m,n,i}|\rceil \leq \lceil 0.5 \cdot a \rceil$ elements in $\mathbf{f}_a$ with size of $a$ to flip down and make $|\sum_i \Delta \widehat{\mathbf{W}}^\ell_{m,n,i}| \leq 0.5$. For example, if $\sum_i \Delta \mathbf{W}^\ell_{m,n,i} = 3.2$, we need to flip 3 elements with positive perturbation (rounding up) to negative (rounding down) with $-3.0$ mutation. Then, we have

$$\Delta \widehat{\mathbf{W}}^\ell_{m,n:} = \underset{\Delta \mathbf{W}^\ell_{m,n,:}}{\arg\min} (\sum_i \Delta \mathbf{W}^\ell_{m,n,i})^2 \Rightarrow |\sum_i \Delta \widehat{\mathbf{W}}^\ell_{m,n,i}| = \left|k - |\sum_i \Delta \mathbf{W}^\ell_{m,n,i}|\right|. \tag{29}$$

For each kernel $|\sum_i \Delta \mathbf{W}^\ell_{m,n,i}| \geq 0$, $\sum_i \Delta \widehat{\mathbf{W}}^\ell_{m,n,i}$ has the minimum value $\left|k - |\sum_i \Delta \mathbf{W}^\ell_{m,n,i}|\right| \leq 0.5$. The sufficiency of Eq. (10) has been proven:

$$\Delta \widehat{\mathbf{W}}^\ell_{m,:} = \underset{\Delta \mathbf{W}^\ell_{m,n,:}}{\arg\min} (\sum_i \Delta \mathbf{W}^\ell_{m,n,i})^2 \Rightarrow \forall \Delta \widehat{\mathbf{W}}^\ell_{m,n,:}, \; |\sum_i \Delta \widehat{\mathbf{W}}^\ell_{m,n,i}| \leq r_k = 0.5 \tag{30}$$

When $a = K, b = 0$ and all perturbation $= 0.5$, original $\sum_i \Delta \mathbf{W}^\ell_{m,n,i}$ have the upper bound $0.5 \cdot K$. □

*Proof.* If we flip $k - 1$ or $k + 1$ elements for $n$-th kernel, the $\sum_i \Delta \widehat{\mathbf{W}}^\ell_{m,n,i}$ can reduce to $\left|k - 1 - |\sum_i \Delta \mathbf{W}^\ell_{m,n,i}|\right|$ and $\left|k + 1 - |\sum_i \Delta \mathbf{W}^\ell_{m,n,i}|\right|$, respectively. Obviously,

$$\left|k - 1 - |\sum_i \Delta \mathbf{W}^\ell_{m,n,i}|\right| = \left|\lfloor|\sum_i \Delta \mathbf{W}^\ell_{m,n,i}|\rceil - 1 - |\sum_i \Delta \mathbf{W}^\ell_{m,n,i}|\right| > 0.5, \tag{31}$$

$$\left|k + 1 - |\sum_i \Delta \mathbf{W}^\ell_{m,n,i}|\right| = \left|\lfloor|\sum_i \Delta \mathbf{W}^\ell_{m,n,i}|\rceil + 1 - |\sum_i \Delta \mathbf{W}^\ell_{m,n,i}|\right| > 0.5. \tag{32}$$

With other numbers $\neq k$, we can also draw the same conclusions. Therefore, when $r_k \leq 0.5$, there is only one value, i.e., the minimum value, with $k$ flipped elements satisfy the Eq. (10). The necessity of Eq. (10) has been proven:

$$\underset{\Delta \mathbf{W}^\ell_{m,n,:}}{\arg\min} (\sum_i \Delta \mathbf{W}^\ell_{m,n,i})^2 \Leftrightarrow \forall \Delta \widehat{\mathbf{W}}^\ell_{m,n,:}, \; |\sum_i \Delta \widehat{\mathbf{W}}^\ell_{m,n,i}| \leq r_k = 0.5 \tag{33}$$

Similarly, we can extend all conclusions to SQuant-C. □

For SQuant, we only consider the flipping operation in one quantization step and select the elements whose sign is the same as $\sum_i \Delta \mathbf{W}_{m,n,i}^{\ell}$ because flipping with more quantization steps (e.g., flip 0.7 to $-1.0$) and the elements with different perturbation signs will cause a more significant perturbation and will violate the Eq. (8). We explain that in the next section.

## B.2 PROOF OF TOK-$k$ PERTURBATION ALGORITHM

*Proof.* We will prove the SQuant-E&K will lead to the top-$k$ algorithm. We have the composition SQuant-E and SQuant-K optimization objective for $n$-th kernel,

$$\underset{\Delta \mathbf{W}_{m,n,:}^{\ell}}{\arg\min} \quad \sum_i (\Delta \mathbf{W}_{m,n,i}^{\ell})^2 + (\sum_i \Delta \mathbf{W}_{m,n,i}^{\ell})^2, \tag{34}$$

which is the first two items of Eq. (8). Without loss of generality, we assume $e = \sum_i \Delta \mathbf{W}_{m,n,i}^{\ell} > 0$, then SQuant needs to flip $k = \lfloor e \rceil$ elements with perturbation $> 0$ to transform $\sum_i \Delta \mathbf{W}_{m,n,i}^{\ell}$ to $e - k$ and is still constant $e - k$ regardless of which $k$ elements are. Therefore, the $k$ elements are only determined by the first item of Eq. (34), $\sum_i (\Delta \mathbf{W}_{m,n,i}^{\ell})^2$. We denote $\mathbf{f}$ as the index set of the $k$ flipped elements and the original perturbation $\mathbf{O} = \Delta \mathbf{W}_{m,n,:}^{\ell}$ for $n$-th kernel. Therefore, $\mathbf{O}_j > 0$, $j \in \mathbf{f}$. Substituting $\mathbf{f}$ and $\mathbf{O}$ in Eq. (34), we have the optimization objective for $\mathbf{f}$ after flipping $k$ elements,

$$\underset{\mathbf{f}}{\arg\min} \quad \sum_t (|\mathbf{O}_t|)^2 + \sum_j (1 - |\mathbf{O}_j|)^2 + (e-k)^2, \quad t \notin \mathbf{f}, \ j \in \mathbf{f}, \ \mathbf{O}_j > 0 \tag{35}$$

$$= \underset{\mathbf{f}}{\arg\min} \quad \sum_i (|\mathbf{O}_i|)^2 - \sum_j (|\mathbf{O}_j|)^2 + \sum_j (1 - |\mathbf{O}_j|)^2, \quad j \in \mathbf{f}, \ \mathbf{O}_j > 0 \tag{36}$$

$$= \underset{\mathbf{f}}{\arg\min} \quad \sum_j [(1 - |\mathbf{O}_j|)^2 - |\mathbf{O}_j|^2], \quad j \in \mathbf{f}, \ \mathbf{O}_j > 0 \tag{37}$$

$$= \underset{\mathbf{f}}{\arg\min} \quad \sum_j (1 - 2|\mathbf{O}_j|), \quad j \in \mathbf{f}, \ \mathbf{O}_j > 0 \tag{38}$$

$$= \underset{\mathbf{f}}{\arg\max} \quad \sum_j (|\mathbf{O}_j|), \quad j \in \mathbf{f}, \ \mathbf{O}_j > 0. \tag{39}$$

Therefore, the Eq. (39) is essentially the top-$k$ perturbation algorithm. We can easily extend the top-$k$ algorithm in SQuant-C and design the perturbation update algorithm in B.3.

$\square$

## B.3 PERTURBATION UPDATE ALGORITHM

SQuant-K initializes all rounded elements as flip candidates. After SQuant-K, we update the flip candidates for SQuant-C as shown in algorithm 4 based on the insight of top-$k$ perturbation algorithm ( B.2).

**Over SQuant** First we define the situation of $k > |e|$ as "Over SQuant" (line 6). For example, if we have a kernel with $e = +1.6$, we need to SQuant it to $-0.4$ to satisfy in $(-0.5, 0.5]$ with flipping $k = 2$ elements $\{2.6, 2.7\}$ to $\{2, 2\}$. Obviously, when SQuant-C needs this kernel to calibrate, the last element 2.7 should be the first and the only candidate (line 7,8) to flip back to the original rounded number 3 to make the $e = +0.6$, due to it has the largest element perturbation in the $k$ fliped elements and the smallest element perturbation $(|-0.3| < 0.5)$ after it flips back. It is vice versa for $e < 0$.

**Under SQuant** For "Under SQuant" (line 9), we need to make the first un-flipped element as the flip candidate (line 10, 11) for SQuant-C, and will lead the kernel to "Over SQuant" with absolute kernel perturbation in $(0.5, 1.0)$ when SQuant-C flips this element of the kernel.

Finally, each kernel has only one candidate flip element for SQuant-C to satisfy Eq. (8). In practice, it is easy to fuse the perturbation update algorithm with the flip algorithm without extra overhead.

## B.4 COMPLEXITY ANALYSIS

The original optimization problem described by Eq. (4) is NP-hard with $O(M \cdot 2^{NK})$. Based on the SQuant approximation, the new optimization objective is to minimize CASE whose complexity is

---

**Algorithm 4:** Perturbation Update Algorithm.

---

**Input:** Weight perturbation $\mathbf{p}$;
**Output:** Updated weight perturbation $\mathbf{p}$;

1 **def** `UpdatePerturbation(`$\mathbf{p}$`):`
2      $e = \sum_i \mathbf{p}_i$ `// Accumulated perturbation (signed CASE).`
3      $\mathbf{p}[e \cdot \mathbf{p} < 0] = 0$ `// Disable Elements/kernels with different sign from` $e$`.`
4      $k = \lfloor |e| \rceil$ `// Flip` $k$ `elements/kernels based on the CASE.`
5      $\mathbf{f} = \text{TopK}(|\mathbf{p}|, k).\text{indices}$ `// Indices of` $k$ `largest perturbation.`
6      **if** $k > |e|$ **then** `// Over SQuant.`
7          $i = \mathbf{f}_k$ `// The` $k$`-th (last) element of` $\mathbf{f}$`.`
8          $v = \mathbf{p}_i$ `//` $0.5 \leq |v| < 1.0$
9      **else** `// Under SQuant.`
10          $i = \text{TopK}(|\mathbf{p}|, k+1).\text{indices}[k+1]$ `// The` $(k+1)$`-th largest element of` $\mathbf{p}$`.`
11          $v = \mathbf{p}_i$ `//` $|v| \leq 0.5$
12      $\mathbf{p} = 0$ `// Disable all elements.`
13      $\mathbf{p}_i = v$ `// The only flip element candidate of the kernel for SQuant-C.`
14      **return** $\mathbf{p}$

---

$O(M \cdot 2^N)$ for SQuant-C, $O(MN \cdot 2^K)$ for SQuant-K and $O(MNK)$ for SQuant-E. SQuant is optimized to a top-$k$ algorithm with a significant complexity reduction to $O(n \cdot log(k))$ for each sub-problem. Our experiments show that after SQuant-K pre-optimization, SQuant-C only requires a tiny top-$k$ number, such as $k = 32$, to satisfy all cases. For a $3 \times 3$ kernel with 9 elements, SQuant-K only needs $k = 4$ because the kernel CASE is always $\leq 0.5K = 4.5$. Finally, their complexity can reduce to linear, $O(M \cdot N \cdot 5)$ for SQuant-C and $O(MN \cdot 9 \cdot 2)$ for SQuant-K.

