# OpenReview forum: "SQuant: On-the-Fly Data-Free Quantization via Diagonal Hessian Approximation"
_ICLR.cc/2022/Conference — ICLR 2022 Poster_

### Official Review · Reviewer_xprS · 2021-10-26

**Correctness:** 2
**Technical Novelty And Significance:** 3
**Empirical Novelty And Significance:** 3
**Recommendation:** 6
**Confidence:** 4

**Main Review:**

Strength: The main strength of this paper is the introduction of multi-scale approximation of Hessian matrix. Due to the scalability issue, it is impossible to store the entire matrix. Traditional methods usually only keep track of the diagonal terms. Inspired by matrix decomposition ideas, this paper adopt a diagonal + block-wise diagonal +  low rank approximation, which is a good idea. Experiment results also show that the matrix decomposition idea work well.

Weakness: I think there are some flaws in the derivation of the optimization objective.

1. In formula (2), H^{W_m^l} = l_m x^l (x^l)^T. But from formula (3) to formula (4), E[H^{W_m^l}] is replaced by E[x^l (x^l)^T], where l_m disappears. Why this step hold true?
2. In formula (7), the objective is \sum_{n, i} e_{n, i} (\Delta W_{m,n,i}^l)^2 + \sum_n k_n \Delta W_{m,n,:}^l J_k \Delta W_{m,n,:}^l^T + c_m \Delta W_{m,:}^l J_NK W_{m,:}^l^T, but in formula (8), the coefficients e_{n, i}, k_n and c_m disappear. I understand that all the coefficients are positive from Appendix A.2., but as long as they are not all equal, the objective should be different without these coefficients. I don't quite understand how we get (8) from (7).

**Summary Of The Paper:**

In this paper, the authors propose a data-free quantization algorithm of deep neural networks called SQuant. The main idea of SQuant is to decompose the Hessian-based optimization objective into three components: element-wise, kernel-wise and channel-wise components. In order to jointly optimize these three objective functions, a constrained absolute sum of error (CASE) is studied and a progressive algorithm is used. Experiment results show that SQuant algorithm is able to keep higher accuracy with the same number of bits, compared to several baseline algorithms.

**Summary Of The Review:**

The motivation to decompose the Hessian matrix into diagonal, block-wise diagonal and low-rank components is the crucial contribution of this paper. Experiment results show that the multi-scale optimization objective lead to good performance after quantization. I don't follow some steps in the derivations of the optimization objective. This paper will be better if the authors can illustrate these steps well.

---

> ### Author Response · Authors · 2021-11-19
> **Response to Reviewer xprS**
>
> We thank the reviewer for the constructive feedback and address the raised concerns as below.
>
> 1. **In formula (2), $H^{W_m^l} = l_m x^l (x^l)^T$. But from formula (3) to formula (4), $E[H^{W_m^l}]$ is replaced by $E[x^l (x^l)^T]$, where $l_m$ disappears. Why this step hold true?**
>
>     The transformation from Equation (3) to Equation (4) omits $l_m$ based on an assumption: $l_m$ is a positive constant and independent of the input data samples. In fact, $l_m$ will not influence the optimization results of each output channel because a constant factor will not change the result of $argmin$ in Equation (3). This method is wildly used in many quantization and pruning researches, such as AdaRound$^{[1]}$, BRECQ$^{[2]}$, and hessian-aware pruning$^{[3]}$.
>
> 2. **In formula (7), the objective is $\sum_{n, i} e_{n, i}$ ($\Delta {W_{m,n,i}^l)}^2$ + $\sum_n k_n \Delta W_{m,n,:}^l J_k \Delta {W_{m,n,:}^l}^T + c_m \Delta W_{m,:}^l J_{NK} {W_{m,:}^l}^T$, but in formula (8), the coefficients $e_{n, i}, k_n$ and $c_m$ disappear. I understand that all the coefficients are positive from Appendix A.2., but as long as they are not all equal, the objective should be different without these coefficients. I don't quite understand how we get (8) from (7).}**
>
>     We agree with the reviewers that Equation (7) and (8)/(9) are not equal. We approximate  Equation (7) to Equation (8)/(9) to enable fast data-free quantization.
>     The approximation error is insignificant as our comprehensive results have shown the high accuracy of the final quantized model in Tables 1 and 2 of the manuscript.
>     The intuition behind the approximation is that we use an iterative process which progressively reduces each term ($\sum_{n,i}e_{n,i} {\Delta W^l_{m,n,i}}^2$ , $\sum_n k_n \Delta W^l_{m,n,:}J_K{\Delta W^l_{m,n,:}}^T$, and $c_m \Delta W^l_{m,:}J_{NK}{\Delta W^l_{m,:}}^T$) of Equation (7). Because each term's coefficient ($e_{n,i}$, $k_n$, and $c_m$) is positive, the reduction of each term would generally lead to the reduction of the precise objective in Equation (7). In **Appendix A.3 of the manuscript revision**, we also provide an empirical analysis of the approximation error between Equation (7) and (8)/(9).
>
> [1] Nagel, Markus, et al. "Up or down? adaptive rounding for post-training quantization." International Conference on Machine Learning. PMLR, 2020.
>
> [2] Li, Yuhang, et al. "BRECQ: Pushing the Limit of Post-Training Quantization by Block Reconstruction." International Conference on Learning Representations. 2020.
>
> [3] Yu, Shixing, et al. "Hessian-aware pruning and optimal neural implant." arXiv preprint arXiv:2101.08940 (2021).

---

### Official Review · Reviewer_hK2m · 2021-11-01

**Correctness:** 3
**Technical Novelty And Significance:** 3
**Empirical Novelty And Significance:** 3
**Recommendation:** 6
**Confidence:** 4

**Main Review:**

This paper is clearly written and well-motivated. After a successful approximation of the Hessian-based approach, the authors discuss how to minimize the necessity of activation distribution information in Section 3 (assuming that input feature maps auto-correlate with each other in a similar way). The experimental results are impressive in Section 4.

(1) What are the limitations of this work? The quality of the proposed method would depend on the validity of the assumptions to ignore activation distribution. If a summary of limitations and assumptions of input distribution is provided, it would be helpful to understand why this work is only limited to certain CNN models as written in the paper. Since Transformers and MLPs are increasingly utilized for the area that has been dominated by CNN models, it would be necessary to describe which dataset or model architecture is best performed by the proposed method.

(2) The flipping approach is limited to up or down with a step size of 1. Is there any chance to improve the accuracy further if we increase the step size (i.e., allowing wider flipping approaches)?

(3) Flipping approach has been introduced to a few previous PTQ techniques (while such techniques usually assume that a calibration set is provided). How close is the proposed work to those previous flipping-based PTQ methods in terms of model accuracy? If the difference is marginal, does it mean that investigating input data distribution is inherently less important for CNN models? Is there any chance such a difference (between data-free PTQ and calibration-based PTQ) can be larger depending on the characteristics of the dataset? Since all experimental data in this paper is given for ImageNet only, this paper might present an impression that the proposed method might be optimized for ImageNet-like datasets.

**Summary Of The Paper:**

This paper proposes a new data-free quantization method that does not require back-propagation nor fine-tuning. The key idea is adopting Hessian-based optimization that can be decomposed into three parts (SQuant-E, K, and C) corresponding to the three dimensions of the weight tensor (using a few approximations, such as cross-layer independence to simplify the optimization). Then, instead of MSE, the authors introduce CASE (constrained ASE) of weight perturbation. The experimental results show that the proposed DFQ method outperforms even GDFQ that is basically a kind of QAT. The proposed technique is especially useful for a low-bit quantization.

**Summary Of The Review:**

This paper introduces a few reasonable assumptions to alleviate the efforts to estimate the input data distribution. Successful approximations lead to improved model accuracy even with a very small number of bits to represent weights.

---

> ### Author Response · Authors · 2021-11-19
> **Response to Reviewer hK2m**
>
> We thank the reviewer for the constructive feedback and address the raised concerns as below.
>
> 1. **(1). What are the limitations of this work? (2). Transformers and MLPs models ...**
>     - (1). We think that the first limitation of our work is the relatively large approximation error in SQuant-C. SQuant-K has a pretty low approximate error for convolution. However, even though SQuant-C can still improve the accuracy, it has higher errors. As the reviewer pointed out, "if the input distribution is provided'', we can reduce the approximation error and achieve better accuracy but with a more complex optimization space. We need to design a different optimization method to tackle new issues introduced by the data-driven approach.
>
>     - (2). For models like Transformers and MLPs that have no convolutions, thus, they can only be optimized by SQuant-C and not SQuant-K. In addition, Transformer-based models lack batch normalization (BN) to narrow the distribution range. As a result, even the data-driven method AdaRound$^{[1]}$ can only quantize the BERT with 8-bit, and a few quantized activations are 16-bit precision. Moreover, the lack of BN leads the other data-free methods to be unavailable, such as GDFQ, and DSG, which adopt the BN-based distribution information matching approach. We believe that these are interesting problems. We would like to solve these issues in our future work.
>
> 2. **Is there any chance to improve the accuracy further if we increase the step size?**
>
>     For SQuant, more than one quantization step will not improve the accuracy. We mentioned this issue in Appendix B.1. Based on Equation (9), we only consider the flipping operation in one quantization step because more quantization steps will cause a more significant perturbation and violate the first term of Equation (9), i.e., MSE about weight quantization. And one quantization step can provide enough flipping elements proven in Appendix B.2.
>
>     Under the data-free promise, one quantization step is the only solution for SQuant. We design experiments on ResNet18 with ImageNet under 4-bit Weight-Only Quantization. We implement SQuant-E&K with different constraint $r_e$. For example, $r_e = 1.4$ means that we allow 2.3 flipping to 1.0 with |-1.3| < 1.4 flipped perturbation over two quantization steps. And $r_e = 0.5$ is exactly rounding method without SQuant.
>
>     We adopt two types of perturbation order. Ascending is the original algorithm of the manuscript. Descending order prefers the largest perturbation. The ascending results are stable when $r_e \geq 0.9$ because one quantization step can provide enough elements to flip. For descending, $r_e = 0.6$ cannot provide enough elements to flip, and $r_e = 0.7$ has the best accuracy. When $r_e > 0.7$, the accuracy degrades with the increase of $r_e$. All descending results are lower than ascending ones, indicating the small flipped perturbation is better for accuracy. As a result, we only set one quantization step for the SQuant.
>
>     | $r_e$ | Ascending (Min First) | Descending (Max First))|
>     |-|-|-|
>     |  0.5 | 48.150    | 48.150
>     |  0.6 | 67.360    | 67.140
>     |  0.7 | 68.052    | 67.910
>     |  0.9 | 68.076    | 67.590
>     |  1.0 | 68.076    | 67.340
>     |  1.2 | 68.076    | 67.086
>     |  1.4 | 68.076    | 66.338
>
> 3. **How close is the proposed work to those previous flipping-based PTQ methods in terms of model accuracy?**
>
>     We compare the SOTA PTQ methods on ResNet18. SQuant has similar accuracy to the PTQ methods with 4-bit quantization. However, 3-bit SQuant has a lower accuracy due to the low resolution, where it needs more hard effort to fine-tune weights. SQuant directly optimizes the perturbation and prunes most of the optimization search space. We think input data and distribution still take an essential role in the lower-bit (< 4-bit) quantization.
>
>     |bit|SQuant|AdaRound|AdaQuant|BRECQ|
>     |-|-|-|-|-|
>     |Source|71.47|71.08|71.08|71.08|
>     |W4A32|69.75|68.71|68.82|70.70|
>     |W3A32|60.78|68.07|58.12|69.81|
>
> 4. **Is there any chance such a difference can be larger depending on the characteristics of the dataset?**
>
>     Empirically, we think the quantization effectiveness depends on the network architectures instead of datasets. As we explain in the first answer of this comment, batch normalization significantly impacts the result, including data-driven PTQ and SQuant. Meanwhile, the absence of convolution will make SQuant-K not applicable. We also provide results on the ResNet18/20 architecture with different datasets. SQuant works well on 5 datasets. But, it is still hard for us to conclude the relationship between the dataset's characteristics and quantization.
>
>     |bit|ImageNet|Cifar10|Cifar100|CUB|SVHN|
>     |-|-|-|-|-|-|
>     |Source|71.47|94.02|70.38|76.67|96.57|
>     |W4A32|69.75|93.76|69.11|76.18|96.40|
>     |W3A32|60.78|91.26|63.50|73.85|95.75|
>
> [1] Nagel, Markus, et al. "Up or down? adaptive rounding for post-training quantization." ICML. 2020.

---

### Official Review · Reviewer_uKzc · 2021-11-02

**Correctness:** 3
**Technical Novelty And Significance:** 4
**Empirical Novelty And Significance:** 3
**Recommendation:** 8
**Confidence:** 3

**Main Review:**

The paper is overall clearly structured and written. The proposed method is well-motivated, and the resultant flipping solution is novel as far as I know. This paper provides a brilliant way to directly use discrete optimization for quantization instead of conventional training with gradient backpropagation. However, the following points still require some more clarifications.
- It is not clear why the first-order term is omitted in equation (1).
- It is not clear why the scaling parameters like e_{n,i}, k_n and c_m
are omitted in equation (7). The solution from the proposed progressive optimization is not the only solution, and other solutions may still depend on these scaling parameters.

**Summary Of The Paper:**

This paper proposes a data-free quantization method based on the second-order Taylor expansion of  loss, where the Hessian matrix is approximated with different levels: element-wise, kernel-wise and channel-wise. The authors progressively determine the quantized weights from element-wise to kernel-wise and then to channel-wise. The derivation and solution of the quantization are novel. Empirical results show that the proposed method outperforms recent data-free methods.



**Summary Of The Review:**

This paper is well written, and the proposed method is novel.

---

> ### Author Response · Authors · 2021-11-19
> **Response to Reviewer uKzc**
>
> We thank the reviewer for the constructive feedback and address the raised concerns as below.
>
> 1. **It is not clear why the first-order term is omitted in equation (1).**
>
>     The first-order item, $\mathbf{g}{\mathbf{W}}$, is the gradient w.r.t. original weight, $\mathbf{W}$. Because a well-trained model has already converged, the first gradient term will be close to $0$ and thus can be safely ignored and omitted. This method is wildly used in prior quantization and pruning researches, such as AdaRound$^{[1]}$, BRECQ$^{[2]}$, and hessian-aware pruning$^{[3]}$.
>
> 2. **It is not clear why the scaling parameters like $e_{n,i}$, $k_n$ and $c_m$ are omitted in equation (7). The solution from the proposed progressive optimization is not the only solution, and other solutions may still depend on these scaling parameters.**
>
>     We agree with the reviewers that Equation (7) and (8)/(9) are not equal. We approximate  Equation (7) to Equation (8)/(9) to enable fast data-free quantization.
>     The approximation error is insignificant as our comprehensive results have shown the high accuracy of the final quantized model in Tables 1 and 2 of the manuscript.
>     The intuition behind the approximation is that we use an iterative process which progressively reduces each term ($\sum_{n,i}e_{n,i} {\Delta W^l_{m,n,i}}^2$ , $\sum_n k_n \Delta W^l_{m,n,:}J_K{\Delta W^l_{m,n,:}}^T$, and $c_m \Delta W^l_{m,:}J_{NK}{\Delta W^l_{m,:}}^T$) of Equation (7). Because each term's coefficient ($e_{n,i}$, $k_n$, and $c_m$) is positive, the reduction of each term would generally lead to the reduction of the precise objective in Equation (7). In **Appendix A.3 of the manuscript revision**, we also provide an empirical analysis of the approximation error between Equation (7) and (8)/(9).
>
> [1] Nagel, Markus, et al. "Up or down? adaptive rounding for post-training quantization." International Conference on Machine Learning. PMLR, 2020.
>
> [2] Li, Yuhang, et al. "BRECQ: Pushing the Limit of Post-Training Quantization by Block Reconstruction." International Conference on Learning Representations. 2020.
>
> [3] Yu, Shixing, et al. "Hessian-aware pruning and optimal neural implant." arXiv preprint arXiv:2101.08940 (2021).

---

### Official Review · Reviewer_cncj · 2021-11-10

**Correctness:** 3
**Technical Novelty And Significance:** 3
**Empirical Novelty And Significance:** 3
**Recommendation:** 6
**Confidence:** 4

**Main Review:**

Strengths:
1. I think the paper is very well written, with good illustrations.
2. The related work section is thorough.
3. The notations in Section 2.1 are necessary and helpful, so as the Algorithm 1 and 2. Otherwise, the notion of blocks, layers, elements, input/output channels, kernels will make the math become super confusing.
4. The experimental results are state-of-the-art, with evaluations on various neural networks.
5. I go through the math and find them generally correct. The decomposition/approximation idea is decent.

Weaknesses:
1. SQuant is for weight and activation quantization. I think the weight quantization part is well supported, but there lacks discussion about activation quantization. Especially, how it affects the accuracy of approximations/Hessian representations used in SQuant.
2. Extra clarifications are needed for Equation 7 to 8. As I understand it, equation 7 to 8 works well for SQuant because of its iterative optimization on the three components. But generally, those two are not equivalent.
3. How is the quantization range (min, max) determined in SQuant? Would SQuant work better if combined with other orthogonal PTQ methods?
4. For completeness, I wonder about 4-bit quantization results of Shufflenet?

Small Issues:
1. In supp A, extra W

======Post Rebuttal======

I think the rebuttal from the authors makes sense and can address my concerns (excluding non-technical issues such as novelty and potential influence). As a result, I maintain my score tending to accept.

**Summary Of The Paper:**

In the manuscript, the authors propose SQuant, which is a data-free quantization method that can apply post-training quantization (PTQ) without any backpropagation.

Specifically, SQuant is taking advantage of approximated Hessian information. Based on the assumptions and deductions in the paper, SQuant tries to optimize constrained absolute sum of error (CASE) instead of MSE.

The authors show many experimental results to validate the effectiveness of SQuant.

**Summary Of The Review:**

I think the manuscript is technically sound with a solid understanding of the PTQ problem. The experimental results are quite good. Since there are still clarifications to be made, and the fact that similar deductions and flipping methods exist in previous works, I recommend a weak acceptance.

---

> ### Author Response · Authors · 2021-11-19
> **Response to Reviewer cncj**
>
> We thank the reviewer for acknowledging the novelty and effectiveness of SQuant, as well as the valuable feedback. We address the raised concerns as below.
>
> 1. **...there lacks discussion about activation quantization. Especially, how it affects the accuracy of approximations/Hessian representations used in SQuant.**
>
>     The activation is generated dynamically and therefore cannot adopt our SQuant optimization. Instead, we embrace the simple rounding method and set the quantization range with the BN method that is also used by previous work DFQ$^{[1]}$ and ACIQ$^{[2]}$. We also mention it in the experiments in Sec. 4.1.
>
>     In practice, it is computationally unaffordable to compute the full Hessian matrix for the activation (feature maps). For the impact of activation quantization on our approximation, we clarify the reviewer's concern in the second part of the following question regarding the approximation analysis.
>
> 2. **Extra clarifications are needed for Equation 7 to 8. As I understand it, Equation 7 to 8 works well for SQuant because of its iterative optimization on the three components. But generally, those two are not equivalent.**
>
>     We agree with the reviewers that Equation (7) and (8)/(9) are not equal. We approximate  Equation (7) to Equation (8)/(9) to enable fast data-free quantization.
>     The approximation error is insignificant as our comprehensive results have shown the high accuracy of the final quantized model in Tables 1 and 2 of the manuscript.
>     The intuition behind the approximation is that, as the reviewer has pointed out, we use an iterative process which progressively reduces each term ($\sum_{n,i}e_{n,i} {\Delta W^l_{m,n,i}}^2$ , $\sum_n k_n \Delta W^l_{m,n,:}J_K{\Delta W^l_{m,n,:}}^T$, and $c_m \Delta W^l_{m,:}J_{NK}{\Delta W^l_{m,:}}^T$) of Equation (7). Because each term's coefficient ($e_{n,i}$, $k_n$, and $c_m$) is positive, the reduction of each term would generally lead to the reduction of the precise objective in Equation (7). In **Appendix A.3 of the manuscript revision**, we also provide an empirical analysis of the approximation error between Equation (7) and (8)/(9).
>
> 	Going back to the reviewer's first question on the impact of activation quantization on our approximation, we compare the approximation precision defined in **Appendix A.3** before and after the activation quantization, and find that the activation quantization actually improves our approximation precision.
>     We think it is because that the quantization on activation plays a clipping effect that eventually limits the coefficients of $e_{n,i}$, $k_n$, and $c_m$. We believe that the study of the approximation is a promising direction and leave it to our next study.
>
> 3. **How is the quantization range (min, max) determined in SQuant? Would SQuant work better if combined with other orthogonal PTQ methods?**
>
>     For all our results, we set the max and min of the tensor for the SQuant **weight** quantization range without any clipping. We implement the search algorithm with minimum MSE$^{[2]}$ for the clipping quantization range with ResNet18 on ImageNet. The results are shown below. Obviously, it will improve the accuracy by combining other orthogonal PTQ methods. SQuant is compatible with further optimizations, such as bias correction$^{[1]}$.
>     |ResNet18|Squant|Squant+MSE|
>     |-|-|-|
>     |Source|71.47|71.47|
>     |W5A32|71.19|71.26|
>     |W4A32|69.75|70.12|
>     |W3A32|60.78|63.08|
>
> 4. **For completeness, I wonder about 4-bit quantization results of Shufflenet?**
>
>     The ShuffleNet results are shown below. Since the ShuffleNet expands convolution without activation (ReLu), its activation has a wider range with negative numbers. Therefore, quantization is bounded at the signed activation.
>
>     |ShuffleNet|Weight&Activation|Weight-only|
>     |-|-|-|
>     |Source| 65.07|65.07|
>     |6-bit| 60.25|64.39|
>     |4-bit| 16.34|51.68|
>
> 5. **Small Issues.**
>
>    Thanks for the correction. We have fixed this typo in the revision.
>
> [1].Nagel, Markus, et al. "Data-free quantization through weight equalization and bias correction." Proceedings of the IEEE/CVF International Conference on Computer Vision. 2019.
>
> [2].Banner, Ron, Yury Nahshan, and Daniel Soudry. "Post-training 4-bit quantization of convolutional networks for rapid-deployment." Proceedings of the 33rd International Conference on Neural Information Processing Systems. 2019.

---

### Decision · Program_Chairs · 2022-01-20

**Decision:**

Accept (Poster)

**Comment:**

The authors propose a data-free quantization method that can be applied post-training quantization without backpropagation.  The method takes advantage of approximate Hessian information in a certain scalable approximate way. Based on the assumptions and deductions in the paper, SQuant tries to optimize constrained absolute sum of error (CASE) instead of MSE.  There are good empirical results showing the effectiveness of the method, and the paper is well written, and the method should be of broad interest.